# Lay People Training in CPR and in the Use of an Automated External Defibrillator, and Its Social Impact: A Community Health Study

**DOI:** 10.3390/ijerph16162870

**Published:** 2019-08-11

**Authors:** Felipe Villalobos, Albert Del Pozo, Cristina Rey-Reñones, Ester Granado-Font, David Sabaté-Lissner, Carme Poblet-Calaf, Josep Basora, Antoni Castro, Gemma Flores-Mateo

**Affiliations:** 1Research Support Unit Tarragona-Reus, Fundació Institut Universitari per a la Recerca a l’Atenció Primària de Salut Jordi Gol i Gurina (IDIAPJGol), Reus 43202, Spain; 2Research Group in Primary Care Research Technologies (TICS-AP, Fundació Institut Universitari per a la recerca a l’Atenció Primària de Salut Jordi Gol i Gurina (IDIAPJGol), Reus 43202, Spain; 3Primary Care Centre Falset, Tarragona Regional Management, Catalan Institute of Health, Tarragona 43730, Spain; 4Nursing Department. University Rovira i Virgili. Tarragona 43002, Spain; 5Primary Care Centre Horts de Miró, Tarragona Regional Management, Catalan Institute of Health, Reus 43204, Spain; 6Primary Care Centre CUAP, Tarragona Regional Management, Catalan Institute of Health, Reus 43202, Spain; 7Faculty of Medicine and Health Sciences. Universitat Rovira i Virgili, Reus 43201, Spain; 8Internal Medicine Department, Sant Joan de Reus University Hospital, Reus 43204, Spain; 9Analysis and Quality Unit, Health and Social Network Santa Tecla, Tarragona 43003, Spain

**Keywords:** out-of-hospital cardiac arrest, cardiopulmonary resuscitation, automatic external defibrillator, community health study

## Abstract

Out-of-hospital cardiac arrest (OHCA) mortality remains high. The best survival rates are achieved when trained people provide OHCA victims with cardiopulmonary resuscitation (CPR); however, it is estimated that only 25% of victims receive CPR. This community health study aims to evaluate the effectiveness of a training programme in basic CPR and in the use of an automatic external defibrillator (AED) on knowledge and skills for lay people, and its social impact. The training courses were based on Catalan Council of Resuscitation guidelines. Data were collected on sociodemographic characteristics, evaluation of knowledge and practical skills at baseline and at the end of the training courses, and also on the social impact of the programme. A total of 36 training courses with 482 participants were carried out, and most participants achieved a qualification of suitable. The mean score in knowledge was 3.1 ± 1.1 at baseline and 3.8 ± 1.2 (*p* = 0.001) at the end of the programme. Participants rated the training courses as very satisfactory, considered the training useful, and felt more qualified to respond to an emergency. This study shows that a high percentage of participants acquired skills in basic CPR and use of an AED, which confirms the usefulness and effectiveness of training courses and its important social impact.

## 1. Introduction

Out-of-hospital cardiac arrest (OHCA) is a significant public health problem. In Europe, over 275,000 cases of OHCA are recorded every year [1]. In Spain, it is estimated that 24,500 cases of OHCA occur yearly, with mortality rates between 79.9% and 84.3% [2,3]. Mortality is mainly associated with the time gap between the cardiac arrest and the arrival of the Emergency Medical Services (EMS) [2,3,4]. It has been observed that OHCA survival is inversely proportional to the time elapsed between OHCA and cardiopulmonary resuscitation (CPR) initiation time. In our region (Catalonia, Spain), the response interval between alerting and arrival of EMS is usually greater than 8 min [5].

Since 70–85% OHCA cases occur at home and 15–25% in public spaces [6], programs that train the lay population in CPR remain crucial. Studies have demonstrated that community education and quality improvement programs to increase bystander and first-responder intervention, including CPR and defibrillation, and access to automatic external defibrillator (AED) are associated with improved outcomes, including survival [7,8,9]. However, most CPR training and guidelines to date have focused on healthcare personnel [10]. Unfortunately, in our population there are scarce data available about CPR training in bystanders (lay population) who are not healthcare personnel. Early CPR by a relative or lay population would achieve maximum effectiveness in the second link of the chain of survival, since CPR maneuvers would be initiated immediately after the loss of consciousness of the OHCA victim [4]. In Europe, 20–70% of OHCA victims receive basic CPR from a relative or a bystander, placing Spain in the penultimate place [11,12].

Several studies report that victims with better survival outcomes and quality of life after having suffered an OHCA are those who received CPR performed by a trained bystander [13,14,15]. Studies conducted in the United States between 2005 and 2010 observed that 27.4% of 31,689 cases of OHCA had better survival outcomes after receiving CPR [16]. In addition, CPR performed before the arrival of EMS more than doubled the 30-day survival rate [17]. 

Due to the high mortality of OHCA and the inadequate response of the lay population to a victim of OHCA, we proposed to conduct a community health intervention, which consisted of training lay people (i.e., individuals who are not healthcare personnel) in basic CPR and use of an AED, with the aim to provide skills in basic CPR and in use of an AED to create a network of qualified citizens able to carry out basic life support in the community.

## 2. Material and Methods

### 2.1. Study Design

Community health study conducted in the city of Reus (Catalonia, Spain; 100,000 inhabitants), which aimed to evaluate the effectiveness of a training programme in basic CPR and in the use of an AED on knowledge and skills for lay people, and its social impact. This study corresponds to Phase 2 of the SmartWatch Project. Study protocol details have been described elsewhere [18].

### 2.2. Participants

Participants were recruited by the research team and primary healthcare professionals, who explained the objectives of the study and offered training courses in basic CPR and use of an AED. People who decided to participate provided contact information (name and phone number). The research coordinator contacted the participants by phone to schedule the date, time, and place of the training course. Participation did not entail any remuneration or economic incentive.

Inclusion criteria were as follows: (1) relatives and/or caregivers of people with heart disease; (2) people who are in daily contact with patients at risk of OHCA (police, firefighters, teachers, shopkeepers, pharmacists, university students, gym instructors, and others); (3) and other adults interested in receiving the training.

According to data obtained from primary care electronic medical records and the database of the city’s hospitals, in Reus 2105 people with heart disease are at risk of OHCA. The training courses aimed to train 430 people (20% of relatives of people with heart disease).

### 2.3. Training 

Knowledge and skills courses in basic CPR and use of an AED were carried out in accordance with Catalan Council of Resuscitation (CCR) guidelines [19], which follow European Resuscitation Guidelines [20]. The courses had a duration of 90 min and were delivered by instructors accredited by the CCR and a member of the research team as support staff. Each group comprised 8–10 participants per instructor and included one training mannequin (BRAYDEN^®^) and one training AED (Saber One^®^). The courses were free, and a certificate of attendance was provided after completion.

The BRAYDEN^®^ training mannequin had an additional function that helped the instructor to visualize what happens with the blood flow to the brain based on the speed and depth of compressions. It consists of a lighting function that will only be lit when both compression depth (over 5 cm) and speed are effective (over 100 per minute).

The AED Saber One^®^ is a training defibrillator with three universal steps (turn on, place the electrodes, and defibrillate). In addition, this training unit has instructions and audio cues with a metronome for the appropriate number and rate of chest compressions (100 per minute) and two ventilations providing easy-to-follow CPR sequence for the participants.

### 2.4. Variables

Sociodemographic characteristics were collected at baseline: date of birth, gender, and social class (British Registrar General questionnaire [21]), which uses three categories: upper (I–II), middle (III_N_–III_M_), and lower (IV–V). 

At baseline and at the end of the training courses, an evaluation assessed the knowledge in basic CPR and AED use. An adaptation of four questions of the questionnaire proposed by the CCR was used: “Evaluation Test for Basic Life Support and Automatic Defibrillation (BLS-AED)” [22] (Figure 1). Each correct answer scored one point, and the results ranged from 0 to 4.

At the end of the training courses, the instructor evaluated the competence of each participant, in accordance with “Practical evaluation. BLS + AED sequence”, proposed by the CCR [23]. Each item scored 1 point: (a) check scene; (b) assess level of consciousness; (c) open airway; (d) check spontaneous breathing; (e) ask for help; (f) chest compressions (quality); (g) frequency of compressions; (h) two ventilations (quality); (i) alternate compressions/ventilations. A trainee received the qualification of suitable when eight of the nine items evaluated were correctly performed. The training mannequin and AED were used to for the practice and for results measurement.

All participants were asked to respond to a course satisfaction questionnaire adapted by members of the research team, which contained three items: contents; methodology/organization; and course instructors. Score was on a scale of 1 (total dissatisfaction) to 5 (total satisfaction).

### 2.5. Social Impact

Six months after the end of the training courses, the social impact of training in basic CPR and AED use was evaluated. The members of the research team sent a text message to all participants with a link to an online survey validated for the Spanish population [24].

### 2.6. Statistical Analysis

The quantitative variables were expressed as mean and standard deviation, and the categorical variables as percentages. The *χ*^2^ test was used to compare categorical variables in different groups. Unpaired Student’s *t*-test was used to compare continuous variables and the paired Student’s *t*-test was used to compare values between different time-points (baseline and end of the training courses) for continuous variables, while the McNemar test was used for the categorical variables.

Statistical significance was set at a *p* value < 0.05. SPSS for Windows Version 22.0 (IBM Corp. Released 2013. IBM SPSS Statistics for Windows, Version 22.0. Armonk, NY, USA) was used for the analysis.

## 3. Results

### 3.1. Sociodemographic Characteristics

From September 2017 to October 2018, 36 training courses where 482 participants were trained in CPR and AED use were conducted (Figure 2). A total of 50.8% trained participants were women; mean age was 45.7 (±2.5) years; 73.5% participants were in active employment and 48.7% were middle class; the group with the highest representation was the general population (60.8%) (Table 1).

### 3.2. Training Evaluation

After evaluating knowledge and practical skills, 419 participants were considered suitable to perform CPR and AED use. The instructors took into account the additional function of the training mannequin (explained above) for practical skills measure of the participants. 

In relation to the evaluation of knowledge in basic CPR and AED use, at the end of the courses knowledge had improved (from 3.17 ± 1.2 to 3.80 ± 1.1 points, *p* = 0.001) and the percentage of participants who responded correctly to each question of the questionnaire increased (Figure 3). In the analysis of knowledge, the firefighters were the group with more correct answers at the beginning and end of the training (*p* < 0.05) (Table 2).

Table 3 shows the results of the training course satisfaction questionnaire. Most participants were satisfied with the training courses, they expressed intention to apply the knowledge learned, and considered necessary to transmit what they had learned to others.

### 3.3. Social Impact

Table 4 reports the evaluation of the social impact of training in basic CPR and AED use. Of 411 participants contacted by text message, only 207 answered the survey (50.36%). Up to 89.9% of surveyed participants reported knowing the meaning of OHCA, 86.6% would be able to identify a victim of OHCA, and a high percentage of participants would recognize a public access AED device. In addition, 28.5% participants reported having witnessed an OHCA on at least one occasion, and of these more than half (52.5%) had tried to perform basic CPR. All participants considered important for the lay population to be trained in basic CPR, in order to assist any victim of OHCA.

When asked for their opinion regarding training in basic CPR and AED use at the end of the survey, most respondents agreed on the following: (a) training courses should be introduced to pupils in primary and secondary schools; (b) the general population should be informed on the location and availability of AED devices in the city; and (c) courses should be carried out more frequently to keep the training up to date.

## 4. Discussion

This community health study evaluated knowledge and skills training in basic CPR and AED use in the lay population. The results showed that a significant percentage of participants acquired sufficient skills in basic CPR and AED use, confirming the usefulness and effectiveness of these courses.

One of the strengths of the study is that the programme was based on Catalan Council of Resuscitation recommendations [19], which follow European Resuscitation Guidelines [20], and allow standardization of knowledge in the lay population. In addition, each instructor was in charge of only 8-10 participants, and mannequins with simulated cases were used for practicing. Several studies indicate that in order to acquire the appropriate skills, CPR maneuvers should be repeatedly applied on mannequins in small groups until the execution becomes automatic [25,26,27,28].

This intervention programme trained 482 participants, more than the initial 430 anticipated, which corresponded to 20% relatives of patients with heart disease registered in the city. CPR international guidelines underscore that any person who is unresponsive and does not breathe normally should be recognized as a victim of OHCA, and once the victim is identified, the EMS should be immediately activated, and witnesses should begin CPR maneuvers until the arrival of the AED and healthcare professionals. Within these guidelines, community health programs of CPR training that educate the lay population remain essential, with special attention to the populations at risk [20,29]. 

Our results show that 86.9% of the trained participants acquired adequate CPR skills. Similar results were observed in a study conducted in the Spanish general population, where 87.2% participants were adequately prepared [26]. In contrast, a study conducted in Spanish high-school students reported lower percentages (58%) [30]. In other countries, similar percentages of participants of various lay population groups achieved sufficient training in CPR and AED use [25,27,31,32].

In our courses, older people with musculoskeletal problems were unable to practice on mannequins and did not attain sufficient CPR skills. A recent study agreed that physical limitation, lack of confidence, and having to practice in large groups were barriers to CPR training in the older population (over 55 years) [33]. Although this group is considered more at risk of OHCA than potential witness, it is essential that they also acquire CPR skills.

At the end of the programme, CPR knowledge increased, meaning that most participants understood the frequency of compressions, compression/ventilation ratios and use of the AED. These results are consistent with other studies conducted in the Spanish population. For instance, high-school students [26] and students from the nursing and physiotherapy degrees [34] showed an increase of knowledge regarding CPR at the end of the training, and university students improved skills in AED use [35]. These studies had longer duration times than ours (20 h in 8 sessions, 4 h and 120 min, respectively). In this respect, our results show that even with limited time and resources, 90 min of CPR training would significantly increase dissemination in the community.

Statistically significant differences were found when comparing the results on knowledge between groups, with firefighters scoring highest at the end of the training. Firefighters, together with the EMS and the police, are usually amongst the first responders to OHCA victims, and consequently CPR and use of the AED is part of their professional training. The inclusion of firefighters in an active network of trained volunteers would increase the survival rate of OHCA in urban and rural areas [36,37,38].

Community health studies indicate the social impact of interventions in the community. In this case, training in basic CPR and AED use in the lay population is expected to have a significant social impact, since a high percentage of trained participants learned to identify cardiac arrest and public access AED devices. In contrast, a recent survey conducted in northern Spain that explored knowledge and attitude with regard to OHCA and CPR techniques, reported that 73% of respondents knew what cardiac arrest was, but only 45% declared that if they witnessed one they would be able to identify it, and only 60% knew how to identify a public access AED device [24]. Similar results were observed in surveys conducted in other European countries [17,39]. 

Future strategies for CPR training and AED use should take into account the suggestions made by a high percentage of participants, namely to introduce training in primary and secondary schools, to update skills regularly, and to disseminate the knowledge of public access AED devices in the community. Several studies report that CPR training in schools is key to the dissemination of CPR in communities, adding students as potential resuscitators of OHCA victims [40,41]. 

Few studies evaluate the effect of the dissemination of public access AED devices. A recent study conducted in Japan observed that the increase in the use of public access AED devices by bystanders was associated with an increase in the number of OHCA survivors, with fewer neurological complications [42]. It is therefore essential to implement public health strategies to disseminate easily accessible AED devices in the community.

## 5. Conclusions

This training programme improved knowledge and skills in basic CPR and in the use of an AED and had a high social impact at community level. We think that it is very important to evaluate the level of knowledge and skills in CPR and use of an AED. Training in basic CPR and use of an AED is a key element of the chain of survival for OHCA, early activation of emergency medical services, immediate bystander provision of CPR, and rapid defibrillation could improve survival outcomes and quality of life for OHCA victims. 

## Figures and Tables

**Figure 1 ijerph-16-02870-f001:**
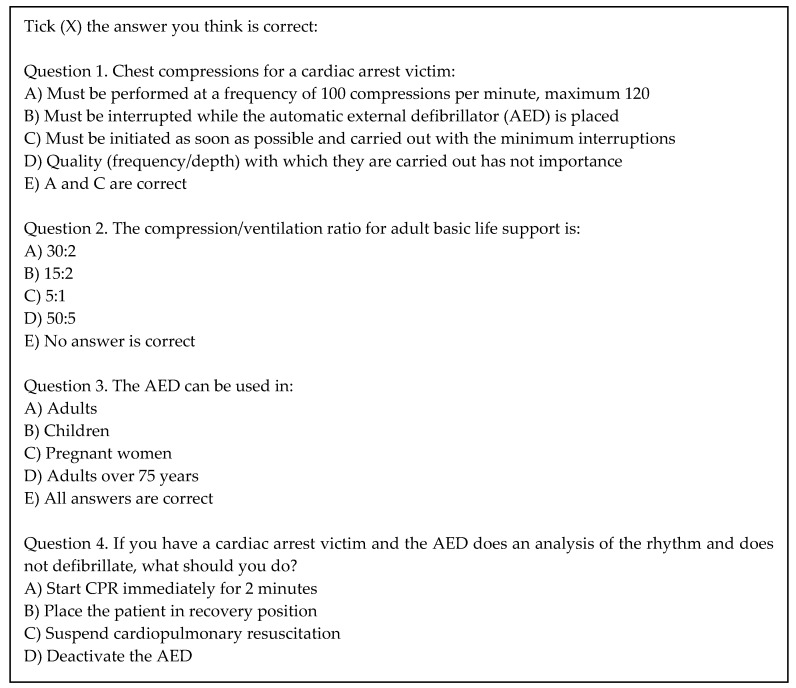
Questionnaire for assessing the knowledge in basic CPR and AED use.

**Figure 2 ijerph-16-02870-f002:**
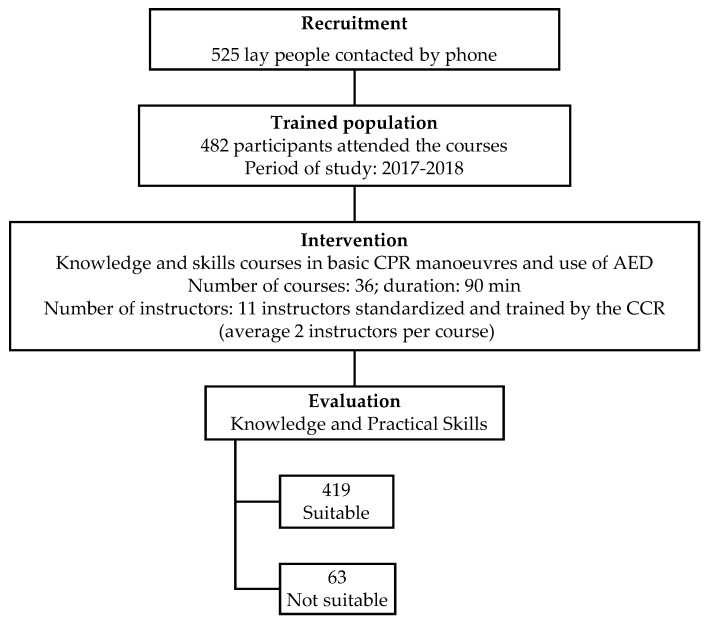
Study flowchart.

**Figure 3 ijerph-16-02870-f003:**
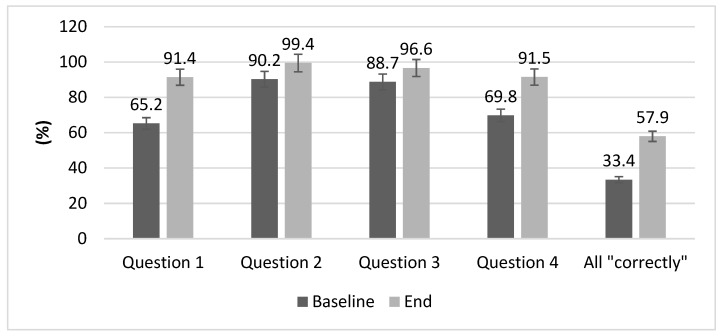
Percentages of participants who responded correctly to the evaluation questionnaire at baseline and at the end of the training. Values expressed in percentage. All questions: *p* < 0.05 between different time-points (baseline and at the end of the training courses).

**Table 1 ijerph-16-02870-t001:** Sociodemographic characteristics of participants.

	Participants (*n* = 482) %
**Age (years) ***	45.7 (2.8)
**Women**	50.8
**Employment status**	
Currently working	73.5
Unemployed	5.8
Retired	9.5
Housewife	3.4
Student	5.8
Disability	1.3
**Social class**	
Lower class (IV–VI)	14.1
Middle class (III_N_–III_M_)	48.7
Upper class (I–II)	37.2
**Group**	
General population	60.8
Catalan Police	13.9
Firefighters	9.8
National Police	5.4
Gym instructors	6.4
Teaching staff	3.1
University students	0.6

* Values expressed as mean and (SD).

**Table 2 ijerph-16-02870-t002:** Percentage of participants who responded correctly to the evaluation questionnaire at baseline and at the end of the training courses by groups.

Group	Question 1		Question 2		Question 3		Question 4	
	Baseline	End	Baseline	End	Baseline	End	Baseline	End
General population *	60.6	93.5	88	99	88.9	97.5	67.7	92.1
Catalan police *	51.1	85.7	95.3	100	90.5	100	69.4	78.8
Firefighters *	93.3	100	100	100	91.5	97.6	88.9	100
National police *	56.5	56.2	78.3	100	73.9	60	65	93.3
Gym instructors *	74.1	91.7	93.5	100	90	100	63.3	92.3
Teaching staff *	78.6	93.3	75	100	93.3	100	68.3	100
University students *	66.7	100	100	100	66.7	100	66.7	100

Values expressed in percentages. * *p* < 0.05 for each question between different time-points (baseline and at the end of the training courses).

**Table 3 ijerph-16-02870-t003:** Training course satisfaction questionnaire.

	Participants (*n* = 361)
**Contents**	
The topics were addressed correctly	4.60 (0.49)
**Methodology/Organization**	
The duration of the training course was appropriate	4.64 (0.49)
**Course instructors**	
The environmental conditions (classroom, furniture, resources used) were adequate to facilitate the training process	4.63 (0.51)
The instructor handled the topics well	4.73 (0.45)
**Appreciation**	
The instructor motivated the participants	4.75 (0.43)
The training course has been useful	4.77 (0.42)

Values expressed in mean and (SD). Rank 1–5.

**Table 4 ijerph-16-02870-t004:** Social impact of training in basic cardiopulmonary resuscitation (CPR) and automated external defibrillator (AED) use.

	Participants (*n* = 207) %
Do you know the meaning of OHCA?	89.9
Would you know how to identify a victim of OHCA?	86.6
Would you be able to recognize a public access AED device?	82.1
Have you ever witnessed an OHCA?	28.5
Have you ever tried to perform basic CPR?	15.9
Considers important knowledge in basic CPR	100

OHCA = out-of-hospital cardiac arrest. AED = automated external defibrillator, CPR = cardiopulmonary resuscitation.

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
