# Peer review of "Lay People Training in CPR and in the Use of an Automated External Defibrillator, and Its Social Impact: A Community Health Study"

_ijerph, 2019, doi:10.3390/ijerph16162870_

Round 1
Reviewer 1 Report
It is an interesting study. Especially as regards the social impact. That is its main value. In the rest of the contributions it is quite repetitive or obvious.
-General features:
the study does not contribute anything that is not known ...... they are taught and learned !!! It also only evaluates knowledge and satisfaction. Quote the sequence but not to develop. The most relevant study is the type of training that is "very brief".
The best, the focus on social impact ...
Important, to check the bibliography. There are articles about "training in CPR" in Spain in this same magazine, of this last year and that nevertheless do not appear.
-Specific:
The questionnaire with 4 questions seems very sensitive and specific.
Talk about "skills" but never measure them. That doll gives you information about compression quality (that's "skills").
How many are the trainers? I think it can be a limitation and justify the differences between groups.
There is a bibliography about brief training and training with "lay people" that he does not quote.
In the discussion he talks about statistical inference and should check it in results and include the p in the tables.
You must provide more information about the doll and the DESA.
The conspiracy has nothing to do with the study nor responds to the objective. For me the goal is very unspecific.
Author Response
7th August, 2019
Dear Reviewer 1
Thank you for your positive response to our manuscript. We have read your comments and we thank them for their constructive and helpful suggestions. We have taken into account all the suggestions and have made the appropriate changes in the revised version of the text.
It is an interesting study. Especially as regards the social impact. That is its main value. In the rest of the contributions it is quite repetitive or obvious.
-General features:
the study does not contribute anything that is not known ...... they are taught and learned !!! It also only evaluates knowledge and satisfaction. Quote the sequence but not to develop. The most relevant study is the type of training that is "very brief".
The best, the focus on social impact ...
Important, to check the bibliography. There are articles about "training in CPR" in Spain in this same magazine, of this last year and that nevertheless do not appear.
We consider that our community health study contributes to the scientific knowledge on the effectiveness of a brief CPR training programme and the use of an ADE in non-health professionals, since most of the training programmes are aimed at health professionals and in based that a high percentage of general population is witnesses of an OHCA. The knowledge and practical skills that participants were provided had a social impact at the community level, we think that every community health study should include the social impact measure as an outcome for further public health strategies.
-Specific:
Point: The questionnaire with 4 questions seems very sensitive and specific.
Answer: The questionnaire is a valided in our population by the “Catalan Council of Resuscitation” and these 4 question are very sensitive and specific.
Point: Talk about "skills" but never measure them. That doll gives you information about compression quality (that's "skills").
Answer: Yes, training mannequin gave us information about compression quality, and instructors took into account the additional function of the training mannequin for practical skills measure of the participants. This has been added and corrected in the revised text.
Point: How many are the trainers? I think it can be a limitation and justify the differences between groups.
Answer: The number of instructors were 11 instructors in total, and all of them were standardized and trained by the “Catalan Council of Resuscitation”. In average 2 instructor per course. This has been added and corrected in the revised text.
Point: There is a bibliography about brief training and training with "lay people" that he does not quote.
Answer: Thank you for the suggestion. This has been added in the discussion section of the revised text.
Point: In the discussion he talks about statistical inference and should check it in results and include the p in the tables.
Answer: Yes, there were statistical inference in our results, we have added the p value in figure 3 and table 2. For each question p value was < 0.05 between different time-points (baseline and at the end of the training courses).
Point: You must provide more information about the doll and the DESA.
Answer: Thank you for the suggestion. This has been provided in the methods section of the revised text.
Point: The conspiracy has nothing to do with the study nor responds to the objective. For me the goal is very unspecific.
Answer: Thank you for the suggestion. This has been re-written in a clear way to explain the objective of the study and the goals achieved.
Cristina Rey-Reñones
Felipe Villalobos Martínez
Research Support Unit Tarragona-Reus
Camí de Riudoms 53-55 43202 Reus (Tarragona), Spain
Tel: +34 977778515
Reviewer 2 Report
This interesting work is important for OHCA and CPR. This work provided the benefit of survival rate if training people perform CPR for OHCA victims. This study evaluated a training program for lay people in basic CPR and in the use of an automatic external 25 defibrillator (AED), and its social impact. This study shows that a high percentage of participants acquired skills in basic CPR and use of an AED. Therefore, the study provided information about training in basic CPR and use of an AED is an important part of the chain of survival for OHCA. This is important for future healthcare system and the policy of government.
Author Response
7th July, 2019
Dear Reviewr 2
Thank you for your positive response to our manuscript.
This interesting work is important for OHCA and CPR. This work provided the benefit of survival rate if training people perform CPR for OHCA victims. This study evaluated a training program for lay people in basic CPR and in the use of an automatic external 25 defibrillator (AED), and its social impact. This study shows that a high percentage of participants acquired skills in basic CPR and use of an AED. Therefore, the study provided information about training in basic CPR and use of an AED is an important part of the chain of survival for OHCA. This is important for future healthcare system and the policy of government.
Cristina Rey-Reñones
Felipe Villalobos Martínez
Research Support Unit Tarragona-Reus
Camí de Riudoms 53-55 43202 Reus (Tarragona), Spain
Tel: +34 977778515
Round 2
Reviewer 1 Report
The authors have endeavored to improve the version of the manuscript.
The current version is ready to be accepted.